# Complex Tasks, Delegation, and Flexibility: What Role for Engagement and Shared Leadership?

António Luis Dionísio [1], Andreia Dionísio [1,2], Maria José Sousa [3] and Ana Moreira [4,*]

1 Center for Advanced Studies in Management and Economics, University of Évora, 7004-516 Évora, Portugal
2 Management Department, University of Évora, 7004-516 Évora, Portugal
3 Department of Political Sciences and Public Policies, Universitary Institute of Lisbon, 1649-026 Lisbon, Portugal
4 School of Psychology, ISPA—Instituto Universitário, 1149-041 Lisboa, Portugal
* Correspondence: amoreira@ispa.pt

**Abstract:** Task complexity is one of the main factors studied by academics and is at the center of leaders' concerns. It is related to delegation and flexibility, which are enhanced by workers' engagement and depend on how leadership is shared. In this context, this research was carried out with the main objective of testing the serial mediating effect of engagement and shared leadership on the relationship between task complexity and flexibility and the relationship between task complexity and delegation. The sample for this study consists of 368 participants, all of whom work in organizations based in Portugal. The results indicate that task complexity is positively and significantly associated with engagement, shared leadership, flexibility, and delegation. Shared leadership has a positive and significant association with flexibility and delegation. The serial mediating effect of engagement and shared leadership on the relationship between task complexity and flexibility was confirmed. The serial mediating effect of engagement and shared leadership on the relationship between task complexity and delegation was not confirmed.

**Keywords:** task complexity; engagement; shared leadership; flexibility; delegation

## 1. Introduction

Today, organizations have the challenge of dealing with an evolving society and a highly diverse labor market, where there are people with different characteristics and behaviors from different generations and cultural backgrounds. In this environment, emerging team-based work structures are designed to meet the requirements and respond to the changes that promote innovation in different types of work (D'Innocenzo et al. 2016).

To address these challenges and to ensure their sustainability, organizations adopt different approaches, creating interactive and dynamic influence processes among groups to lead to shared goals (Hoch and Dulebohn 2017), in the hope that this will contribute to improved performance (Bamford and Griffin 2008; Sangeetha and Kumaran 2018). In this innovative management approach, work teams have influenced organizational structures by playing leadership roles and acting effectively (Martin et al. 2018; Sweeney et al. 2019). Innovative approaches emerge in this context where shared leadership predominates instead of the traditional vertical leadership structure (Martin et al. 2018; Sweeney et al. 2019; Zhu et al. 2018).

With these working trends, talented members who can perform multidisciplinary roles and are available for collective leadership behaviors, involvement, and commitment to teams emerge (Zhu et al. 2018). Therefore, a shift occurs from a vertical command structure to a shared leadership process among team members (Carson et al. 2007; D'Innocenzo et al. 2016; Martin et al. 2018; Zhou et al. 2015; Zhu et al. 2018).

Shared leadership is a phenomenon that emerges within teams over time (Nicolaides et al. 2014). This notion is shared by authors who find various successful interactions

between team members (Carson et al. 2007; Sweeney et al. 2019; Zhou et al. 2017). This study aims to review the literature on the impact of shared leadership on teamwork performance. The existing literature examines shared leadership as a concept and its impact on team performance without specifying or targeting studies on specific areas or activities (Zhu et al. 2018; Sweeney et al. 2019).

On the other hand, the wealth of recent empirical studies has continually demonstrated the positive effects that the influence of shared leadership has on team performance. The theory has shown that not all teams with shared leadership cultures may be equally effective (Kukenberger and D'Innocenzo 2019). This process may be related to the implications associated with autonomous teams' performance, recruitment, and socialization (Siangchokyoo and Klinger 2022). However, the literature reinforces the importance of responsibilities and team homogeneity in the role of shared leadership that, rather than allocating this role to a single leader, develops collective influence in this practice. Thus, although studies indicate the existence of some patterns in the teams' environment that may allow the influence of individuals to commit to this dynamic, there is some consensus on the role of mediators in influencing this practice in effective teams.

One of the key factors studied by academics and at the center of leaders' concerns is the task's difficulty. He is associated with delegation and flexibility, both of which are strengthened by workers' commitment and are reliant on how leadership is shared. In this situation, the main goal is to investigate whether engagement and shared leadership are the mechanisms that explain both the relationship between task complexity and flexibility and between task complexity and delegation.

In addition, the main intention of this study is to present a model that can contribute to the evolution of knowledge, creating a favorable environment for the performance of small teams.

### 1.1. Task Complexity and Engagement

In the work context, one of the relevant characteristics is task complexity. Task complexity refers to the cognitive demands present in any professional occupation (Schaubroeck et al. 1994) and the amount of information the employee processes in performing his/her tasks (Schmidt et al. 2008). Task complexity fosters employees' high perception of self-efficacy (Judge et al. 2000), enhancing their intrinsic motivation (Oldham and Cummings 1996). According to Delaney and Royal (2017), intrinsic motivation is the main predictor of engagement because when employees feel highly motivated, they do more than they are asked.

Engagement is considered as employees' involvement, enthusiasm, commitment, passion, absorption, effort, focus, and energy when performing their tasks (Schaufeli 2013). As predictors of engagement, we have work resources (autonomy, supervision/coaching, and performance feedback) and personal resources (self-efficacy) (Bakker and Demerouti 2008). In addition, according to these authors, resources become more relevant to high work demands. According to Bakker and Demerouti (2008), job and personal resources are the main predictors of engagement, and these resources are even more relevant in a highly demanding work context. In addition, for Sohrabizadeh and Sayfouri (2014), job characteristics are associated with work engagement as antecedents of it. The following hypothesis was thus formulated:

**Hypothesis 1.** *Task complexity is significantly and positively associated with engagement levels.*

### 1.2. Task Complexity and Shared Leadership

In the organizational context, leadership has been intensely explored and discussed. Fiedler (1996), besides considering that leadership processes are highly complex, also considers that leadership is an interaction between the leader and what is translated from leadership in his/her practice.

As for shared leadership, the literature states that the sharing process can occur in such a way that groups work together in time and place, or it can occur over time,

where its members emerge as informal leaders serially or alternately in the leadership role (Lord et al. 2017; Pearce 2004; Zhu et al. 2018).

Day et al. (2004) suggested that shared leadership constitutes a condition of mutual influence among team members and their interactions, enabling team and organizational performance. However, Carson et al. (2007) argued that shared leadership arises with individual team members engaging in activities that influence the team and other team members in management, motivation, and support areas. DeRue et al. (2015) suggested a complex and adaptive process involving several significant and subsequent interactions.

Furthermore, D'Innocenzo et al. (2016) identify two roles: the members who take the lead and provide guidance, motivation, and support to teammates and the role of the following members who receive guidance, motivation, and support. For Serban and Roberts (2016), shared leadership development is associated with the team environment provided by cohesion in task performance and a common goal. Second, according to Sangeetha and Kumaran (2018) and Lin and Peng (2010), cohesion influences performance through sharing trust and common perspectives in a favorable environment among members. Mutual and collective influence is an essential characteristic of shared leadership, leading to theoretical reasoning about how leadership thrives on reciprocal influence among team members who must know when to lead and follow (Nassif 2019; Nicolaides et al. 2014; Zhou et al. 2017).

While the concern of researchers is to find a position where shared leadership is evident, it ultimately derives from more traditional leaders. However, despite being a diminishing influence between leader and followers, shared leadership is more than just an articulation between members (D'Innocenzo et al. 2016; Storm and Scheepers 2019; Zhou et al. 2017); that is, the individuals who make up teams adopt this way of life in most organizations (Han and Beyerlein 2016). In this sense, Han and Beyerlein (2016) highlight the importance of the preparation and integration of individuals in these processes and the need to learn and use increasingly new types of collaboration tools.

Consequently, shared leadership has different definitions, many derived from how it is operationalized or aggregated and many focused on leadership influence. For Mehra et al. (2006) and Sivasubramaniam et al. (2002), it is a shared and distributed phenomenon where it is possible to formally find multiple leaders, either within or outside the group. Carson et al. (2007) identify the distribution of leadership influence among various team members. Wang et al. (2014) define shared leadership as an emergent team property of mutual influence and shared responsibility, where they lead (each other toward the achievement of goals).

Some definitions of shared leadership focus on the number of people involved in leadership activities to distinguish shared leadership from more traditional leadership (D'Innocenzo et al. 2016; Nicolaides et al. 2014; Zhu et al. 2018). Pearce and Conger (2003) consider it a dynamic and interactive process of mutual influence within groups to achieve organizational goals through reciprocal leadership. As for shared leadership, the literature states that the sharing process can occur in such a way that groups work together in time and place, or it can occur over time, when its members emerge as informal leaders serially or alternately in the leadership role (Lord et al. 2017; Pearce 2004; Zhu et al. 2018).

From the perspective of Rose et al. (2021), the best team members can reflect on complex tasks in teamwork by adopting leadership behaviors appropriate to individual talent and situational demands. For Pearce (2004), tasks of high interdependence, complexity, and creativity are suitable for shared leadership. The following hypothesis was thus formulated:

**Hypothesis 2.** *Task complexity is positively and significantly associated with shared leadership.*

### 1.3. Task Complexity, Flexibility, and Delegation

Flexibility can be defined as companies' abilities to adapt to new circumstances, to new competitive realities, to innovate and implement technology, ready to respond quickly to market demands (Atkinson 1988).

Flexibility has thus become a central aspect of organizations in managing the work and qualifications of employees. It is even a critical point in current management that seeks

to find the desirable combination between production factors and the ability to generate value through their HR (Sequeira 2008). There are several types of flexibility, but we will focus on functional flexibility, which can be considered one of the preferred strategies of organizations because it allows the organization to reduce the levels of division and fragmentation of work, enabling the development of multidisciplinary tasks and employee versatility (Thompson et al. 2007).

In this sense, a new employee profile emerges of who can perform different tasks, act, think, act, plan strategically, and innovate in solving new problems (Sequeira 2008). The multifunctional employee has certain differentiating characteristics that allow him/her to perform multiple functions or tasks.

Delegation can be considered a leadership technique that helps employees by allowing them to participate in decision making, increasing their self-esteem and encouraging better communication and relationships in the workgroup (Ugoani 2020). Delegation is strongly linked to empowerment, as this concept is also linked to self-efficacy. Employees feel empowered and responsible when they participate in decision making on issues that may affect their performance. Delegation fosters feelings of trust and recognition in employees, improving the relationships between leaders and followers (Zhang et al. 2017).

In a study by Eggertsson and Le Borgne (2006), these authors conclude that the more skills are required to perform a task, the more desirable it becomes to delegate. They concluded that task complexity is positively associated with delegation.

In turn, Barrow (1976) find that task complexity is associated with flexibility, with employees becoming more flexible when faced with more complex tasks. The following hypothesis was formulated:

**Hypothesis 3.** *Task complexity has a positive and significant association with flexibility and delegation.*

### 1.4. Shared Leadership, Flexibility, and Delegation

The interest in shared leadership emerged in parallel with the need for organizations to adopt projects based on more agile and responsive teams, created by the increased complexity of functions and the rapidly changing nature of work. Ideas such as participatory decision-making, self-leadership, team self-management, training, and knowledge are some of the important scientific contributions in the last four decades (Carson et al. 2007; Pearce and Conger 2003). These rapid changes in work demand great flexibility from teams, which is facilitated by shared leadership.

For Zhang et al. (2017), an authentic leader should promote positive leadership, delegating power and authority to their employees so they have more freedom to work autonomously, leading to greater job satisfaction, organizational commitment, and innovative behaviors. This reasoning led us to formulate the following hypothesis:

**Hypothesis 4.** *Shared leadership has a positive and significant association with flexibility and delegation.*

### 1.5. Serial Mediator Effect

The main predictors of engagement are job and personal resources, which are even more important when the tasks are more demanding and complicated (Bakker and Demerouti 2008). In turn, engagement can facilitate an orientation toward shared leadership (Gautvik-Minker and Skjelbred 2017) and this can foster greater flexibility (Pearce and Conger 2003) and delegation (Zhang et al. 2017). We conclude that engagement and shared leadership are the mechanisms that explain the relationship between task complexity and flexibility and the relationship between task complexity and delegation. This reasoning led us to formulate the following hypotheses:

**Hypothesis 5.** *Engagement and shared leadership have a serial mediating effect on the relationship between task complexity and flexibility.*

**Hypothesis 6.** *Engagement and shared leadership have a serial mediating effect on the relationship between task complexity and delegation.*

To synthesize the hypotheses formulated in this study, a theoretical model was developed in which the associations between the constructs studied are presented (Figure 1).

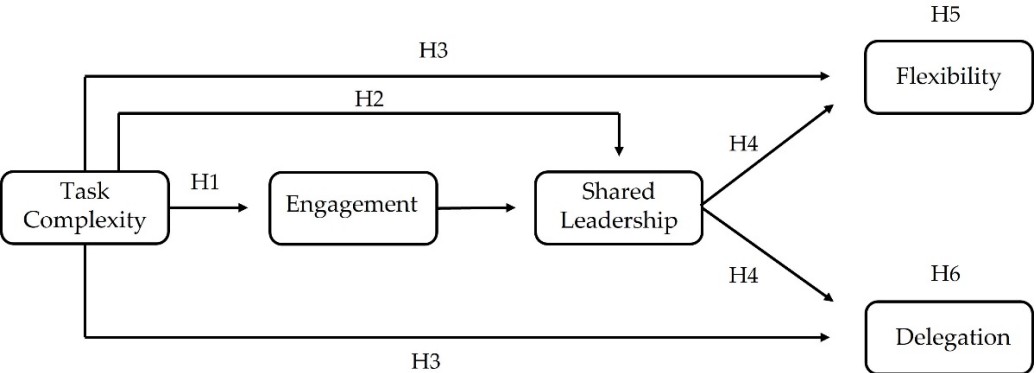

**Figure 1.** Research model.

## 2. Method

### 2.1. The Pre-Test

The development of a questionnaire is subject to specific requirements, which are fundamental for a survey to be conducted with the least possible difficulty and, above all, understood by the respondents (Vilelas 2009; Coutinho 2014). Therefore, a pre-test was conducted to assess the adequacy and understanding of the questions, as well as the clarity of the instructions for completion. This is a process that aims to increase the efficiency and effectiveness of the research itself.

Being a preliminary version, we intended to obtain a sample through individuals from the target public in the universe to be researched. We asked 12 individuals from our target audience to answer the questionnaire and analyze the clarity and ambiguity of the questions, the identification of issues they considered important to the problem, the time it took to complete the questionnaire, and the spelling mistakes they identified. Nine individuals from five different organizations responded.

We collected the suggestions, and, after analyzing them, we proceeded to improve the questionnaire and consequently develop a new version. After the new version was completed, the whole process was repeated, but this time only for seven individuals from four different organizations (all responded). When no more corrections or improvements were identified as needed, the final version was prepared to be applied in the present and future research.

### 2.2. Selection of Sampling Units

The characteristics of the "target" population for this study were employees who are, or have been, involved in the dynamics of work in more agile teams, in person or physically relocated.

The choice of this population would allow greater objectivity in the inputs collected through respondents who develop and promote work environments in line with the research theme. The choice to circumscribe the target population had as its main objective the avoidance of respondents in work dynamics and with functional characteristics different from the objectives under study (Vilelas 2009; Coutinho 2014).

In this sense and considering that the method for data collection relied on the questionnaire survey, an explanatory note was introduced with the objectives and contextualization of the study and the framework of the target audience for which the survey was intended: "The 'target' population of this study are employees of companies that are or have been involved in the dynamics of work in more agile teams in person or physically relocated".

In this way, whenever the respondent did not identify with the theme, he/she could leave the questionnaire, stating that "he/she did not accept to participate in the study".

In addition, in this sense and to be the most assertive in this selection, in the invitations sent to the target population individually, we challenged the respondents to invite two or three people who performed functions related to the study.

### 2.3. Data Collection Procedure

Concerning research, data collection was based on available information using the questionnaire survey research technique (Sampieri et al. 2013). The questionnaire survey (one of the techniques most used by researchers) was chosen because it is one of the most widely used methods in the management area and the one that has greater advantages in terms of cost reduction, a greater probability of data processing, and greater flexibility and error reduction (Babbie 2010; Coutinho 2014; Vilelas 2009). On the other hand, it is widely used to obtain information specifically focused on essential and key aspects to verify previously formulated hypotheses (Barañano 2008). The sampling process was non-probability, convenience, and intentional snowball sampling (Trochim 2000).

In this sense, the questionnaire was built using the Google Forms application associated with a link to make it possible to use via the internet. The questionnaire contained information about the purpose of the study. Participants were asked to be sincere in their answers and to guarantee confidentiality. They were also informed that the individual answers would never be known since the analysis that would later be made would be of all the participants. The questionnaire consisted of sociodemographic questions and five scales (task complexity, engagement, shared leadership, flexibility, and delegation). Data were collected between June and October 2021.

### 2.4. Participants

The sample of this study was composed of 368 participants, working in organizations based in Portugal. As regards to the variable ages, they varied between 21 and 71 years (M = 45; SD = 9.66). Regarding gender, 98.95% of the respondents identified their gender, which translated into 41.3% belonging to the female gender and 57.6% belonging to the male gender. About academic training, the respondents predominantly have higher education, which translates into 81.5% with academic training for a bachelor's degree, master's degree, or doctorate, 14.9% with secondary education, and 3.5% with professional or other education. Regarding the functions that the respondents carry out in the organizations, 72% have middle or senior management positions, 25.3% are operational or assistants, and only 2.7% identified other functions. This shows that the respondents are mainly leaders or managers. Concerning seniority in the job, we found a range of 39 years in the seniority of the respondents' functions, representing an average of 11 years (SD = 8.56).

### 2.5. Data Analysis Procedure

Data were imported into SPSS Statistics 28 for Windows software (IBM Corp., Armonk, NY, USA). The first step was to test the metric qualities of the instruments used in this study. To test their validity, several exploratory factor analyses (EFAs) were performed. This procedure aimed to investigate the correlations between the original variables to estimate the common factors and the structural relationships that link the factors to the variables.

After the EFAs, a confirmatory factor analysis (CFA) was performed using AMOS Graphics 28 for Windows software (IBM Corp., Armonk, NY, USA). A robust maximum-likelihood-type estimation was performed for one-factor and eight-factor models. The procedure was undertaken according to "model generation" logic (Jöreskog and Sörbom 1993), considering interactively the results obtained in the analysis of its fit: for the chi-square ($\chi^2$); for the Tucker–Lewis index (TLI); for the goodness-of-fit index (GFI); for the comparative fit index (CFI); for the root mean square error the approximation (RMSEA); and for the root mean square residual (RMSR). To analyze the other metric qualities of

the instruments used in this study, the SPSS Statistics 28 for Windows software was used. Reliability was then analyzed by calculating Cronbach's alpha for each instrument.

The association between the variables was then studied using Pearson's correlations. Finally, the hypotheses formulated in this study were tested. Hypotheses 1–4 were tested through linear regressions after testing the respective assumptions. To test the mediation model (Hypotheses 5 and 6), we used the PROCESS 4.0 macro developed by Hayes (2013) (Hayes, NY, USA) since it allows us to test a mediation model with multiple mediators operating in series.

### 2.6. Instruments

The instrument for this study was built on items from other instruments by the following authors: Atkinson (1988); Bruccoleri et al. (2019); Carson et al. (2007); Contractor et al. (2012); DeRue et al. (2015); D'Innocenzo et al. (2016); Han et al. (2017); Han and Beyerlein (2016); Hersey and Blanchard (1986); Hoch and Dulebohn (2017); Martin et al. (2018); Mathieu et al. (2015); Morgeson et al. (2010); Nassif (2019); Nicolaides et al. (2014); Schaufeli and Salanova (2007); Storm and Scheepers (2019); Sweeney et al. (2019); Thompson et al. (2007); Zhou et al. (2015); and Zhu et al. (2018) (Appendix A).

All items in this instrument are rated on a 5-point Likert-type rating scale (from 1 "strongly disagree" to 5 "strongly agree").

Task complexity was measured through five items. In the EFA, a KMO value of 0.75 was obtained, which is considered acceptable (Sharma 1996), and Bartlett's test of sphericity was significant at $p < 0.001$. This factor explains 56.24% of the total variability of the scale. Item 1 had to be removed because it had a low factor weight. The confirmatory factor analysis showed that the adjustment indexes were adequate ($\chi^2/gL = 1.61$; GFI = 0.99; CFI = 0.99; TLI = 0.99; RMSEA = 0.041; RMSR = 0.006). A VEM = 0.43 was obtained, a value slightly below 0.50. As for internal consistency, Cronbach's alpha is 0.73 and composite reliability is 0.77.

The engagement was measured through five items, presenting a KMO value of 0.79, and Bartlett's test of sphericity was significant at $p < 0.001$. It was found that the factor structure of this scale is based on one, which explains 58.75% of the total variability of the scale. The confirmatory factor analysis showed that the adjustment indexes were adequate ($\chi^2/gL = 4.34$; GFI = 0.99; CFI = 0.99; TLI = 0.96; RMSEA = 0.090; RMSR = 0.009), which means that participants perceived this scale as being composed of one factor. As for convergent validity, a VEM = 0.49 was obtained. In analyzing the internal consistency of the instrument, it has a Cronbach's alpha of 0.81 and a composite reliability of 0.82.

The shared leadership scale, composed of five items, presents a KMO of 0.66, slightly below the minimum acceptable value (Sharma 1996), and Bartlett's test of sphericity was significant at $p < 0.001$. It was found that the factor structure of this scale is based on a factor of 70.22% of the total variability of the scale. The confirmatory factor analysis showed that the adjustment indexes were adequate ($\chi^2/gL = 3.29$; GFI = 0.98; CFI = 0.97; TLI = 0.93; RMSEA = 0.079; RMSR = 0.013). Convergent validity has a value of 0.58. When analyzing the internal consistency of the shared leadership instrument, it has a Cronbach's alpha value of 0.78 and a composite reliability value of 0.79.

The five items that measure flexibility present a KMO value of 0.85, and Bartlett's test of sphericity was significant at $p < 0.001$. It was found that the factor structure of this scale is based on one factor, which explains 64.13% of the total variability of the scale. The confirmatory factor analysis revealed that the adjustment indexes were appropriate ($\chi^2/gL = 2.72$; GFI = 0.99; CFI = 0.99; TLI = 0.98; RMSEA = 0.068; RMSR = 0.008). Concerning convergent validity, a value of 0.55 was obtained. When analyzing the internal consistency of the flexibility instrument, it has a Cronbach's alpha value of 0.86 and a composite reliability value of 0.86.

The five items that measure delegation present a KMO value of 0.81, and Bartlett's test of sphericity was significant at $p < 0.001$. It was found that the factor structure of this scale is based on one, which explains 54.59 % of the total variability of the scale. The

confirmatory factor analysis was performed, and the adjustment indexes were found to be adequate ($\chi^2$/gL = 2.19; GFI = 0.99; CFI = 0.99; TLI = 0.98; RMSEA = 0.057; RMSR = 0.021). Concerning convergent validity, a VEM = 0.45 was obtained. When analyzing the internal consistency, it presents a Cronbach's alpha of 0.79 and a composite reliability of 0.80.

Concerning the sensitivity of the items and scales, the absolute values of skewness and kurtosis were below 3 and 7, respectively, so they do not grossly violate normality (Kline 1998).

## 3. Results

The first step was to perform descriptive statistics of the variables under study.

### 3.1. Descriptive Statistics of the Variables under Study

To understand the answers given by the participants regarding the variables under study, descriptive statistics were performed.

The participants' answers are all significantly above the central point, which indicates that the participants in this study have a high perception of task complexity, engagement, shared leadership, flexibility, and delegation (Table 1). It should be noted that delegation is the variable with the lowest employee perception, and engagement is the variable with the highest perception.

**Table 1.** Results of Student's *t*-test for one sample.

| Variável | *t* | *p* | Mean | SD |
|---|---|---|---|---|
| Task Complexity | 60.15 *** | <0.001 | 4.37 | 0.44 |
| Engagement | 46.45 *** | <0.001 | 4.31 | 0.54 |
| Shared Leadership | 36.08 *** | <0.001 | 4.22 | 0.65 |
| Flexibility | 48.53 *** | <0.001 | 4.28 | 0.51 |
| Delegation | 18.58 *** | <0.001 | 3.69 | 0.71 |

Note: *** $p < 0.001$.

### 3.2. Correlations

We then tested the association between the variables in the study using Pearson's correlations.

As can be seen in Table 2, all the variables under study are significantly correlated with each other. The strongest association is between task complexity and flexibility (r = 0.58; $p < 0.001$) and the weakest between flexibility and delegation (r = 0.32; $p < 0.001$).

**Table 2.** Association between variables under study.

| | 1 | 2 | 3 | 4 | 5 |
|---|---|---|---|---|---|
| 1. Task Complexity | – | | | | |
| 2. Engagement | 0.46 *** | – | | | |
| 3. Shared Leadership | 0.39 *** | 0.51 *** | – | | |
| 4. Flexibility | 0.58 *** | 0.49 *** | 0.37 *** | – | |
| 5. Delegation | 0.27 *** | 0.43 *** | 0.47 *** | 0.32 *** | – |

Note: *** $p < 0.001$.

### 3.3. General Model

Two models, one-factor and five-factor, were tested. The fit indices of the one-factor model proved to be not adequate ($\chi^2$/gL = 5.67; GFI = 0.67; CFI = 0.62; TLI = 0.58; RMSEA = 0.113; SMRM = 0.064). The fit indices of the five-factor model proved to be adequate ($\chi^2$/gL = 1.81; GFI = 0.92; CFI = 0.95; TLI = 0.94; RMSEA = 0.047; SMRM = 0.031). Thus, the theoretical conceptualization, which determined five variables, adequately represents the observed data.

### 3.4. Hypotheses

The hypotheses formulated in this study were then tested.

**Hypothesis 1.** *Task complexity is significantly and positively associated with engagement levels.*

Hypothesis 1 was tested by performing a simple linear regression.

The results indicate that task complexity has a positive and significant association with engagement ($F_{(1, 366)} = 96.84$; $R^2 = 0.21$; $\beta = 0.46$; $t = 9.84$; $p < 0.001$) (Table 3). The model explains 21% of the variability in engagement. This hypothesis was confirmed.

**Table 3.** Results of simple linear regression (H1).

| Independent Variable | Dependent Variable | F | p | R² | β | t | p |
|---|---|---|---|---|---|---|---|
| Task Complexity | *Engagement* | 96.84 *** | <0.001 | 0.21 | 0.46 *** | 9.84 *** | <0.001 |

Note: *** $p < 0.001$.

**Hypothesis 2.** *Task complexity is positively and significantly associated with shared leadership.*

Hypothesis 2 was tested by performing a simple linear regression.

The results indicate to us that task complexity has a positive and significant association with shared leadership ($F_{(1, 366)} = 65.59$; $R^2 = 0.15$; $\beta = 0.29$; $t = 8.16$; $p < 0.001$) (Table 4). The model explains the variability of shared leadership by 15%. This hypothesis was confirmed.

**Table 4.** Results of simple linear regression (H2).

| Independent Variable | Dependent Variable | F | p | R² | β | t | p |
|---|---|---|---|---|---|---|---|
| Task Complexity | Shared Leadership | 66.59 *** | <0.001 | 0.15 | 0.39*** | 8.16*** | <0.001 |

Note: *** $p < 0.001$.

**Hypothesis 3.** *Task complexity has a positive and significant association with flexibility and delegation.*

Hypothesis 3 was tested by performing two simple linear regressions.

The results indicate to us that task complexity has a positive and significant effect on delegation ($F_{(1, 366)} = 28.61$; $R^2 = 0.07$; $\beta = 0.27$; $t = 5.35$; $p < 0.001$) and flexibility ($F_{(1, 366)} = 189.80$; $R^2 = 0.34$; $\beta = 0.58$; $t = 13.78$; $p < 0.001$) (Table 5). Complexity explains the variability in delegation by 7% and the variability in flexibility by 34%. This hypothesis was corroborated.

**Table 5.** Results of the two simple linear regressions (H3).

| Independent Variable | Dependent Variable | F | p | R² | β | t | p |
|---|---|---|---|---|---|---|---|
| Task Complexity | Delegation | 28.61 *** | <0.001 | 0.07 | 0.27 *** | 5.35 *** | <0.001 |
| | Flexibility | 189.80 *** | <0.001 | 0.34 | 0.58 *** | 13.78 *** | <0.001 |

Note: *** $p < 0.001$.

**Hypothesis 4.** *Shared leadership has a positive and significant association with flexibility and delegation.*

Hypothesis 4 was tested by performing two simple linear regressions.

The results indicate to us that shared leadership has a positive and significant effect on delegation ($F_{(1, 366)} = 102.56$; $R^2 = 0.22$; $\beta = 0.47$; $t = 10.13$; $p < 0.001$) and flexibility

(F (1, 366) = 58.38; $R^2$ = 0.14; β = 0.37; $t$ = 7.64; $p$ < 0.001) (Table 6). Shared leadership explains 22% of the variability in delegation and 14% of the variability in flexibility. This hypothesis was corroborated.

**Table 6.** Results of the two simple linear regressions (H4).

| Independent Variable | Dependent Variable | F | $p$ | $R^2$ | β | $t$ | $p$ |
|---|---|---|---|---|---|---|---|
| Shared Leadership | Delegation | 102.56 *** | <0.001 | 0.22 | 0.47 *** | 10.13 *** | <0.001 |
| | Flexibility | 58.38 *** | <0.001 | 0.14 | 0.37 *** | 7.64 *** | <0.001 |

Note: *** $p$ < 0.001.

**Hypothesis 5.** *Engagement and shared leadership have a serial mediating effect on the relationship between task complexity and flexibility.*

This hypothesis stated that engagement and shared leadership represent a serial indirect effect on the relationship between task complexity and delegation. Specifically, model 1 presents the results of this hypothesis.

As can be seen in Table 7, a significant total indirect effect was observed since the confidence interval did not contain a zero. This indirect effect is divided into three significant indirect effects: the serial indirect effect, the indirect effect in which engagement mediates the relationship between task complexity and delegation, and the indirect effect in which shared leadership mediates the relationship between task complexity and delegation. When analyzing the contrasts, we found that the strongest indirect effect is the one in which affective commitment mediates the relationship between training and intentions to leave the organization. When the mediators were introduced in the regression equation, the direct effect of training on exit intentions ceased to be significant, which leads to the conclusion that we are dealing with a total mediation effect and that this hypothesis was confirmed (Figure 2).

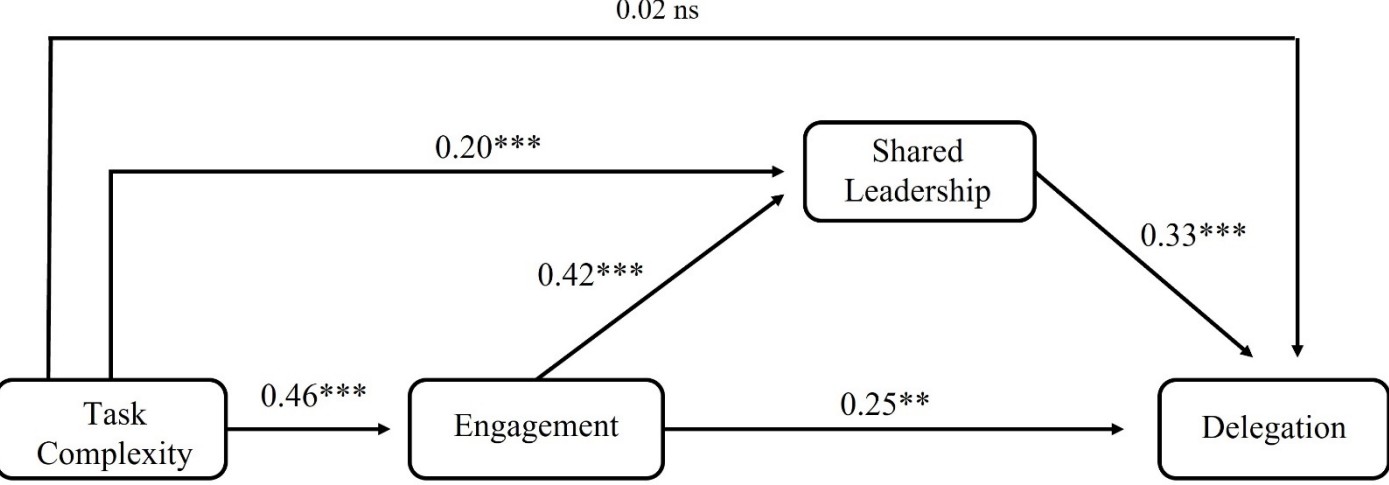

**Figure 2.** Model 1. Notes: ns = no significant; ** $p$ < 0.01; *** $p$ < 0.001.

**Table 7.** Indirect effects of model 1.

| | Indirect Effects | |
| --- | --- | --- |
| | Estimates | 95% Confidence Interval with Bootstrap Correction |
| Model 1 | | |
| Total | 0.24 (0.03) | [0.18; 0.32] |
| Task Complexity → E → D | 0.12 (0.03) | [0.06; 0.18] |
| Task Complexity → SL→ D | 0.07 (0.02) | [0.03; 0.11] |
| Task Complexity → E→ SL → D | 0.06 (0.01) | [0.04; 0.09] |

Notes: total effect task complexity → D = 0.27 (0.08); standard error is in parentheses; E = engagement; SL = shared Leadership; D = delegation.

**Hypothesis 6.** *Engagement and shared leadership have a serial mediating effect on the relationship between task complexity and delegation.*

This hypothesis stated that engagement and shared leadership represent a serial indirect effect on the relationship between task complexity and Flexibility. Specifically, model 2 shows the results of this hypothesis.

As can be seen from Table 8, a significant total indirect effect was observed since the confidence interval did not contain a zero. This indirect effect is divided into three indirect effects, not all of them significant. Only the indirect effect in which engagement mediates the relationship between task complexity and flexibility was significant. The serial spillover effect and the spillover effect in which shared leadership mediates the relationship between task complexity and flexibility were insignificant (Figure 3). This hypothesis was not confirmed.

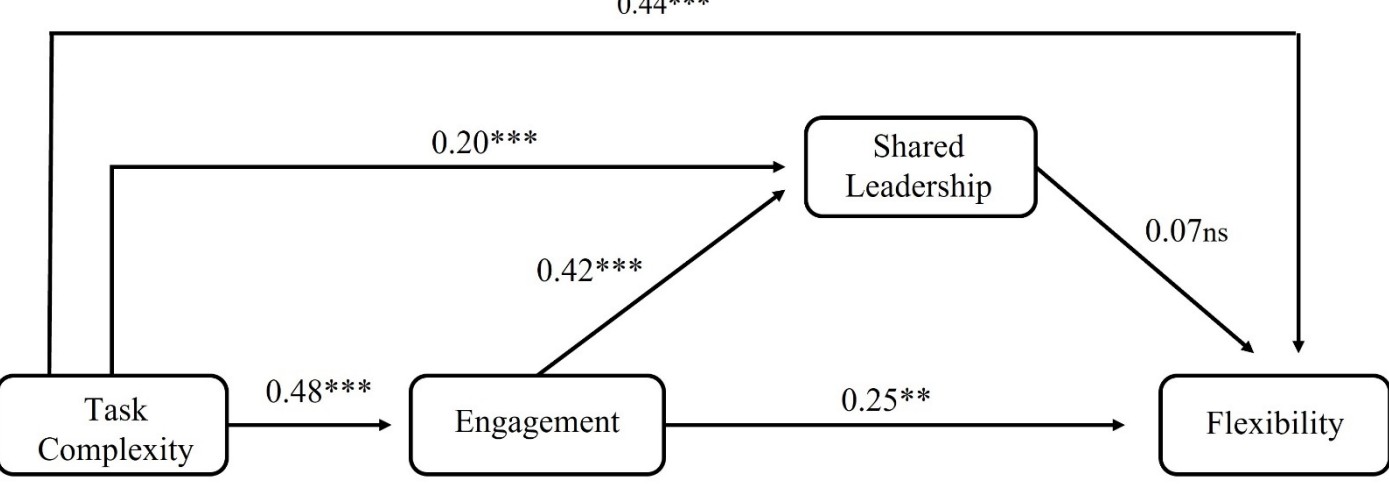

**Figure 3.** Model 2. Notes: ns = no significant; ** *p* < 0.01; *** *p* < 0.001.

**Table 8.** Indirect effects of model 2.

| | Indirect Effects | |
|---|---|---|
| | **Estimates** | **95% Confidence Interval with Bootstrap Correction** |
| Model 1 | | |
| Total | 0.14 (0.04) | [0.08; 0.22] |
| Task Complexity → E → F | 0.11 (0.03) | [0.06; 0.18] |
| Task Complexity → SL→ F | 0.01 (0.01) | [−0.01; 0.04] |
| Task Complexity → E→ SL → F | 0.01 (0.01) | [−0.01; 0.04] |

Notes: total effect task complexity → F = 0.58 (0.05); standard error is in parentheses; E = engagement; SL = shared leadership; F = flexibility.

Finally, Table 9 was elaborated to synthesize the results from the six hypotheses formulated in this study.

**Table 9.** Synthesis of the hypotheses results.

| **Hypothesis** | **Decision** |
|---|---|
| Hypothesis 1: Task complexity is significantly and positively associated with engagement levels. | Supported |
| Hypothesis 2: Task complexity is positively and significantly associated with shared leadership. | Supported |
| Hypothesis 3: Task complexity has a positive and significant association with flexibility and delegation. | Supported |
| Hypothesis 4: Shared leadership has a positive and significant association with flexibility and delegation. | Supported |
| Hypothesis 5: Engagement and shared leadership have a serial mediating effect on the relationship between task complexity and flexibility. | Supported |
| Hypothesis 6: Engagement and shared leadership have a serial mediating effect on the relationship between task complexity and delegation. | Supported |
| Hypothesis 1: Task complexity is significantly and positively associated with engagement levels. | Not supported |

## 4. Discussion

The main objective of this study was to test the serial mediating effect of engagement and shared leadership on the relationship between task complexity and flexibility and the relationship between task complexity and delegation.

In the first place, and as expected, there was a positive and significant association between task complexity and levels of engagement, i.e., the more complex the task, the higher the levels of engagement. These results align with what the literature tells us because, according to Bakker and Demerouti (2008), job and personal resources are the main predictors of engagement, and these resources are even more relevant in a highly demanding work context.

Second, there was a positive and significant association between task complexity and shared leadership, which indicates that leadership behaviors appropriate to individual talent and situational demands should be adopted when faced with a more complex task. These results align with Pearce (2004), who argues that tasks with high interdependence, complexity, and creativity are suitable for shared leadership.

Third, and as expected, there was a positive and significant association between task complexity, flexibility, and delegation. These results align with what Eggertsson and Le Borgne (2006) state, which is that the greater the number of skills required to perform a given task, the more essential it becomes to delegate. As for the association between task complexity and flexibility, in Barrow's (1976) view, employees become more flexible when they must perform more complex tasks. However, it should be noted that the association

between task complexity and flexibility is much stronger than the association between task complexity and delegation. These results are possible since the participants in this study are parts of small teams, which forces them to become more flexible when faced with complex tasks.

Fourth, as hypothesized, shared leadership is positively and significantly associated with flexibility and delegation. From the perspective of Carson et al. (2007), in the last four decades, some of the crucial contributions to science relate to participatory decision-making, self-leadership, team self-management, training, and knowledge, which has led to rapid changes in work, requiring great flexibility, which can be facilitated by shared leadership. Regarding the association between shared leadership and delegation, Zhang et al. (2017) state that authentic leaders should promote positive leadership by delegating power and authority to employees so they have more freedom to work autonomously.

The serial mediating effect of engagement and shared leadership on the relationship between task complexity and delegation was confirmed. These results are in line with what the literature tells us because when teams must perform more demanding and complicated tasks, their engagement levels increase (Bakker and Demerouti 2008), facilitating an orientation toward shared leadership (Gautvik-Minker and Skjelbred 2017) and delegating power and authority to their employees.

Finally, the serial mediating effect of engagement and shared leadership on the relationship between task complexity and flexibility was not proven. This fact may have happened because the relationship between task complexity and flexibility was the strongest, which made it cancel out the other relationships in the regression equation. These results go against what was expected and what the literature tells us.

These results prove the importance of shared leadership to cement workers' commitment, operationalized by delegation and flexibility, depending on the complexity of the task. These results give clues to leaders in the performance of their functions to keep workers committed to the strategic objectives and to the success of the organizations. Concerning academia, it proves that the research conducted in Portuguese organizations obtained results like those of other international studies, solidifying the knowledge on this topic and allowing for new analyses and perceptions on this topic within a culture where the hierarchical distance is high, as is the aversion to uncertainty (Hofstede et al. 1991).

### 4.1. Limitations

This study has some limitations. The first limitation is that it is a cross-sectional study, which did not allow for establishing causal relationships between the variables. To test causal relationships, a longitudinal study would be necessary. The fact that self-report questionnaires were used is another limitation, which may have biased the results. We followed several methodological and statistical recommendations to reduce the impact of common variance (Podsakoff et al. 2003).

Regarding limitations, it is important to refer to our sample not being random and, probably, it is not entirely representative of the population under study. Given this, the results and conclusions cannot be generalized, so our conclusions are mainly focused on the sample under investigation.

### 4.2. Practical Implications

This study's strength is that it shows us that engagement and shared leadership are the mechanisms that explain the relationship between task complexity and delegation. Organizations should invest in small teams when facing demanding and complicated tasks, leading their employees to feel higher levels of engagement (Bakker and Demerouti 2008), promoting the orientation toward shared leadership so that power and authority are delegated to employees so that they have more freedom to work autonomously, leading to higher job satisfaction, greater organizational commitment, and innovative behaviors (Zhang et al. 2017).

## 5. Conclusions

We concluded that our study achieved almost all the proposed objectives, and its conclusions contributed to the advancement of research in organizational behaviors. These conclusions assume even greater importance when it comes to very small teams that must perform complex tasks that require flexibility from the employee and that managers know how to delegate.

Task complexity has a positive and significant association with levels of engagement, shared leadership, delegation, and flexibility.

We found that engagement and shared leadership are the mechanisms that explain the relationship between task complexity and delegation.

Among the proposed objectives, only the serial mediating effect of engagement and shared leadership on the relationship between task complexity and flexibility could not be proved.

This study proved that task complexity enhances employees' engagement (Bakker and Demerouti 2008), promoting an orientation toward shared leadership and delegating power and authority to employees so that they have more freedom to work autonomously (Zhang et al. 2017).

**Author Contributions:** Conceptualization, A.L.D. and A.D.; methodology, A.L.D.; software, A.M.; validation, A.L.D., A.D. and M.J.S.; formal analysis, A.M.; investigation, A.L.D. and A.D.; resources, A.L.D. and A.D.; data curation, A.M.; writing—original draft preparation, A.L.D., A.D. and M.J.S.; writing—review and editing, A.L.D. and A.D.; visualization, A.M. and M.J.S.; supervision, A.D. and M.J.S.; project administration, A.L.D., A.D. and M.J.S.; funding acquisition, A.M. All authors have read and agreed to the published version of the manuscript.

**Funding:** This research received no external funding.

**Institutional Review Board Statement:** Ethical review and approval were waived for this study because all participants, before answering the questionnaire, had to read the informed consent and agree to it. It was the only way they could answer the questionnaire. Participants were informed of the purpose of the study and that the results would be confidential, as individual results would never be known but would only be analyzed in the set of all participants.

**Informed Consent Statement:** Informed consent was obtained from all subjects involved in the study.

**Data Availability Statement:** The data presented in this study are available on request from the corresponding authors. The data are not publicly available since, in their informed consent, participants were informed that the data were confidential and that individual responses would never be known, as data analysis would be of all participants combined.

**Conflicts of Interest:** The authors declare no conflict of interest.

## Appendix A

| Dimension | Indicators\|Key | Scale | Source |
|---|---|---|---|
| I. General Company Data | Sector of activity | 1. Consulting<br>2. Banking/Insurance<br>3. Innovation/Technology<br>4. Services<br>5. Support Business<br>6. Other (Which?) | Barañano (2008)<br>Coutinho (2014)<br>Sampieri et al. (2013) |
|  | Start of activity | Scale |  |
|  | Company typology | 1. Multinational<br>2. SA<br>3. Other (Which?) |  |

| Dimension | Indicators \| Key | Scale | Source |
|---|---|---|---|
| II. Collaboration | To meet my goals, I work daily with my teammates. | 1. Strongly Disagree<br>2. Disagree<br>3. Neither Agree Nor Disagree<br>4. Agree<br>5. Totally Agree | Bruccoleri et al. (2019)<br>Nassif (2019)<br>Zhu et al. (2018)<br>Han and Beyerlein (2016)<br>Nicolaides et al. (2014)<br>Hoch and Dulebohn (2017) |
|  | At least two other members of my team and I, we get along well. |  |  |
|  | My team is made up of members who use collaborative tools. |  |  |
|  | I know what the talents and skills of each of the other members of my team are. |  |  |
|  | I can think of at least two other team members who act as informal leaders in addition to the officially appointed team leaders. |  |  |
|  | The less conflict, the more trust and cohesion, the better the well-being of all my team members. |  |  |
| III. Delegation | The formal leader of my team is able to transfer any authority to informal leaders. | 1. Strongly Disagree<br>2. Disagree<br>3. Neither Agree Nor Disagree<br>4. Agree<br>5. Totally Agree | Bruccoleri et al. (2019)<br>Han et al. (2017)<br>Mathieu et al. (2015)<br>Hoch and Dulebohn (2017)<br>Hersey and Blanchard (1986) |
|  | When major decisions must be made, team members are actively involved in the decision-making process. |  |  |
|  | If a new challenge occurs, the participants' talents and hierarchical position decide the leadership roles. |  |  |
|  | For any operation our team undertakes, several individuals are held accountable for knowledge and decision-making. |  |  |
|  | Because of the way tasks are distributed among team members, the current functions work in a dynamic and interactive process. |  |  |
| IV. Leadership | As a team leader, I am responsible for various tasks and positions. | 1. Strongly Disagree<br>2. Disagree<br>3. Neither Agree Nor Disagree<br>4. Agree<br>5. Totally Agree | Zhu et al. (2018)<br>Martin et al. (2018)<br>D'Innocenzo et al. (2016)<br>Nicolaides et al. (2014)<br>Contractor et al. (2012)<br>DeRue et al. (2015)<br>Morgeson et al. (2010)<br>Carson et al. (2007) |
|  | I feel that my activities involve the other members of the team. |  |  |
|  | The activities I carry out also constitute an orientation (or have a guiding role) for the other members. |  |  |
|  | My team's way of life is to work. |  |  |
|  | Any other member of the team, in my opinion, has the potential to lead. |  |  |

| Dimension | Indicators \| Key | Scale | Source |
|---|---|---|---|
| V. Work Complexity "Talent Management" | When the team "feels" that the work is complex, the probability of success is lower. | 1. Strongly Disagree<br>2. Disagree<br>3. Neither Agree Nor Disagree<br>4. Agree.<br>5. Totally Agree | Storm and Scheepers (2019)<br>Sweeney et al. (2019)<br>Martin et al. (2018)<br>Zhu et al. (2018)<br>Zhou et al. (2015) |
| | Diversity of management skills can improve team performance. | | |
| | I am able to create solutions related to my work. | | |
| | The diversity of skills among team members can improve everyone's performance. | | |
| | A good option for organizations to respond to the rapidly changing nature of work is to adopt more agile teams. | | |
| VI. Culture | Our team members depend on each other to function efficiently and effectively. | 1. Strongly Disagree<br>2. Disagree<br>3. Neither Agree Nor Disagree<br>4. Agree<br>5. Totally Agree | Bruccoleri et al. (2019)<br>Wang et al. (2014)<br>Zhou et al. (2015)<br>Bergman et al. (2012) |
| | When I think of leadership, I imagine a joint purpose to study and create awareness collaboratively. | | |
| | I am confident in my abilities to lead this team. | | |
| | Collective effectiveness is something my whole team relies on. | | |
| | Team structures based on different skill characteristics promote group performance. | | |
| VII. Vision | My team has a clear goal and defined priorities. | 1. Strongly Disagree<br>2. Disagree<br>3. Neither Agree Nor Disagree<br>4. Agree<br>5. Totally Agree | Sweeney et al. (2019)<br>Nassif (2019)<br>Hoch and Dulebohn (2017)<br>Zhou et al. (2017)<br>Han et al. (2017)<br>Mathieu et al. (2015) |
| | I am conscious (aware) of my team's mission and priorities. | | |
| | The mission and priorities of my team are clear to me. | | |
| | My group's leadership positions are based on the needs that arise in connection with our goals. | | |
| | The maturity of the employees (experience) is fundamental for the composition of the best team. | | |

| Dimension | Indicators \| Key | Scale | Source |
|---|---|---|---|
| VIII. Engagement | I am committed with my team to perform multidisciplinary functions. | 1. Strongly Disagree<br>2. Disagree<br>3. Neither Agree Nor Disagree<br>4. Agree<br>5. Totally Agree | Zhu et al. (2018)<br>Wang et al. (2014)<br>Albdour and Altarawneh (2014)<br>Schaufeli and Salanova (2007)<br>Robinson et al. (2004) |
| | I am proud of the work I do. | | |
| | I feel enthusiastic about my work. | | |
| | My enthusiasm for the role I play allows me to be more proactive, more personal initiative and inspiration. | | |
| | I consider my commitment to the organization to be important in achieving significant results with high performance. | | |
| IX. Perceptions on Flexibility | I will take the opportunity to learn more and help my colleagues. | 1. Strongly Disagree<br>2. Disagree<br>3. Neither Agree Nor Disagree<br>4. Agree<br>5. Totally Agree | Thompson et al. (2007)<br>Atkinson (1988) |
| | I believe that employee flexibility is the way to adjust their roles. | | |
| | The company must invest in the flexibility of its employees. | | |
| | These accumulations require continuous learning and development. | | |
| | Teams work best with the flexibility of all their members. | | |
| X. Socio-Demographic Information | Age | Scale | Barañano (2008)<br>Coutinho (2014)<br>Sampieri et al. (2013) |
| | Sex | 1. Woman<br>2. Man<br>3. Prefer not to Identify | |
| | Academic background | 1. Vocational Education<br>2. Secondary Education<br>3. Graduation<br>4. MA<br>5. Doctorate<br>6. Other | |
| | Which of the following best describes your role? | 1. Assistant \| Assistants<br>2. Operational \| Professional<br>3 Manager \| Team Leader<br>4. Director \| Team Manager<br>5. Senior \| Manager<br>6. Executives<br>7. Other (Which?) | |
| | How many years have you been in this position? | Scale | |
| | How many years have you been working in the current organization? | Scale | |

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
