# Peer review of "Complex Tasks, Delegation, and Flexibility: What Role for Engagement and Shared Leadership?"

_socsci, doi:10.3390/socsci11120565_

Round 1
Reviewer 1 Report
Dear author(s). Let me congratulate you for doing research and the drive to publish and contribute to the field. Your manuscript in its current form needs some improvement as well as additional data to properly evaluate your research:
The core of any primary research is the used data and the way it has been collected. In your case, you do not share enough information to support your research design and data collection. The following questions need to be discussed:
1) What was your target participant group? From whom did you want to collect data. Obviously, not every person is suitable as a participant. You lack to provide the trageted profile. From the information shared, it looks like you did not define your traget participant group.
2) How was the google document (survey) shared/ promoted (this relates to the target audience)?
3) You do not define the geographic limitation of your study. From which countries and more important cultures are your participants?
4) This combined leads to the question of sampling.
5) How did you ensure that the particpants understand the complexity of the terminology?
6) Unfortunately, you did not share the questionnaires and how you inquired
a) your variables
b) the connection between the variables
To evaluate your analysis you must first ensure that the data collected (based on your questionnaires) really represents the variables and their dependencies.
7) You do as well not share the participants' resposes (before your analysis).
8) Before analysing dependencies with task complexity, you need to define the dimension of the task complexity - which should have been done for your participants to ensure equal understanding of all participants.
Kind regards
Author Response
Dear Reviewer,
We appreciate your preliminary comments that will complement our work.
Comment 1: What was your target participant group? From whom did you want to collect data. Obviously, not every person is suitable as a participant. You lack to provide the trageted profile. From the information shared, it looks like you did not define your target participant group.
Considering that the method for data collection fell on the survey by questionnaire, an explanatory note was introduced with the objectives and contextualization of the study and framing of the target audience for which the survey was intended: "The "target" population of this study are employees of companies that are or have been involved in the dynamics of work in more agile teams in person or physically relocated. In order to obtain greater quality in the selection, in the invitations sent to the target population individually, we challenged the respondents to invite 2 to 3 people who performed functions related to the study. In this way, whenever the respondent did not identify with the theme, he or she could leave the questionnaire, stating that "he or she did not accept to participate in the study".
Comment 2: How was the google document (survey) shared/ promoted (this relates to the target audience)?
This study's "target" population is composed of employees who are or have been involved in the dynamics of work in more agile teams in person or physically relocated.
The choice of this population allows greater objectivity in the inputs collected through respondents who develop and promote work environments framed with the research theme. The option to circumscribe the target public is to avoid respondents in work dynamics and functional characteristics different from the objectives under study. In this sense, the online questionnaire collected information from June 18 and October 7, 2021. After the closure, the data were exported through Google Forms, as explained above.
Comment 3: You do not define the geographic limitation of your study. From which countries and more important cultures are your participants?
The data were collected in Portugal. We added this in the abstract and in the method (in the sample description).
Comment 4: This combined leads to the question of sampling.
Sample was convenience, purposive, and snowball type (found in the article in the data collection procedure on page 5, lines 232 and 233)
Comment 5: How did you ensure that the particpants understand the complexity of the terminology?
In the introductory note of the questionnaire, a framework clarifying the theme, the target audience, and the study's objectives were carried out. If the respondent was unclear, he/she could simply answer, "I do not wish to participate in this study", and was automatically removed from the study.
Comment 6: Unfortunately, you did not share the questionnaires and how you inquired
Since this is research for a PhD thesis, the questionnaires will be shared by request and will be made public later in the PhD thesis.
- your variables
Independent variable: Task complexity
Dependent Variables: Flexibility and Delegation
Mediating variables: Engagement and Shared Leadership.
- b) the connection between the variables
To evaluate your analysis, you must first ensure that the data collected (based on your questionnaires) really represents the variables and their dependencies.
The answer is given in comment 6a.
Comment 7: You do as well not share the participants' resposes (before your analysis).
Considering that this is research for a PhD thesis, the database will be shared by request.
Comment 8: Before analysing dependencies with task complexity, you need to define the dimension of the task complexity - which should have been done for your participants to ensure equal understanding of all participants.
As task complexity depends on the functions of each respondent, we made the methodological choice of not limiting the concept through its definition in the questionnaire. Therefore, the definition of task complexity was based on different perspectives, different authors and different studies to give it the broadest possible scope.
A table was added at the end of the results to summarize the confirmed and unconfirmed hypotheses. This makes the results clearer.
We hope that we have dealt with yours’s suggestions satisfactorily and made all the adjustments requested, both in form and substance.
Yours sincerely,
On behalf of my co-authors,
Reference added to the manuscript:
- (Hofstede et al., 1991) Hofstede, Gert Jan, Gert Jan Hofstede, and Michael Minkov. 1991. Cultures and Organizations: Software of the Mind. London: McGraw-Hill.
Reviewer 2 Report
Dear authors,
The paper was interesting but it still needed small corrections. The article lacks a clearly formulated scientific problem or problematic questions. Once the scientific problem has been formulated, the results obtained will need to be adjusted accordingly. Scientific problem should be added to the abstract also. The methodological part lacks information on how respondents were selected and found. I don't think the abstract needs to list all the hypotheses. The abstract should describe the importance, novelty of the scientific problem, the essential theoretical model and the obtained conclusions, and not state which hypotheses are proven and which are not. So your title question: what role for engagement and shared leadership?
sincerely
Author Response
Dear Reviewer,
We are very thankful for all your interesting insights.
Comment 1: The paper was interesting but it still needed small corrections.
Thank you very much for all the questions and comments that helped the article become more solid.
Comment 2: The article lacks a clearly formulated scientific problem or problematic questions.
The introduction has been improved by clarifying the scientific problem, which is operationalized in the hypotheses that have been defined.
Comment 3: Once the scientific problem has been formulated, the results obtained will need to be adjusted accordingly. Scientific problem should be added to the abstract also.
The problem will be clarified in the introduction. However, since it is already aligned with the previously defined hypotheses, no reformulation of the results will be necessary. Only a deeper discussion of the results was obtained. The problem has been added in the abstract.
Comment 4: The methodological part lacks information on how respondents were selected and found.
No interviews were conducted. The participants answered a questionnaire in which the sample by convenience, intentional and snowball. It is found in the article in the data collection procedure (page 5, lines 232 and 233)
Comment 5: I don't think the abstract needs to list all the hypotheses.
The hypotheses were removed from the abstract.
Comment 6: The abstract should describe the importance, novelty of the scientific problem, the essential theoretical model and the obtained conclusions, and not state which hypotheses are proven and which are not.
The abstract was changed, the hypotheses were removed, and the scientific problem was clarified.
Comment 7: So your title question: what role for engagement and shared leadership?
The role of engagement and shared leadership is that of a serial mediating effect in the relationship between task complexity and flexibility, and delegation.
A table was added at the end of the results to summarize the confirmed and unconfirmed hypotheses. This makes the results clearer.
We hope that we have dealt with yours’s suggestions satisfactorily and made all the adjustments requested, both in form and substance.
Yours sincerely,
On behalf of my co-authors,
Reference added to the manuscript:
- (Hofstede et al., 1991) Hofstede, Gert Jan, Gert Jan Hofstede, and Michael Minkov. 1991. Cultures and Organizations: Software of the Mind. London: McGraw-Hill.
Round 2
Reviewer 1 Report
Dear author(s).
Thank you for your partially revised version. Your changes are minimalistic and do not address the major insulated to your sample and the survey questions.
Considering your response that the survey questions will be published later with the PhD thesis leads me to advise for a later submission, when you can present all necessary data. Your comment that the survey questions are available on request, well the last revision requested the survey questions.
The additional information that the research sample are companies from Portugal is not sufficient. What sectors do these companies work in, what are the company sizes, etc. are important variables that were not considered.
Non-probability, convenience, and intentional snowball sampling cannot be used if not a specific cluster of participants is supposed to be researched. Your sampling method is not representative as convenience sampling with snow balling leads to a biased group of participants. As you did not mention that you target with your research a specific group of participants, random sampling would have been needed to show a generally applicable result.
Author Response
Dear Reviewer
We appreciate your preliminary comments that will complement our work.
Comment 1: Considering your response that the survey questions will be published later with the PhD thesis leads me to advise for a later submission, when you can present all necessary data. Your comment that the survey questions are available on request, well the last revision requested the survey questions.
After consideration, doctoral student, and advisors, we decided to add the questionnaire to the article. The questionnaire is in Appendix A.
Comment 2: The additional information that the research sample are companies from Portugal is not sufficient. What sectors do these companies work in, what are the company sizes, etc. are important variables that were not considered.
In the method, we have added two subchapters that, we think, answer your comment. The two subchapters are:
2.1. the pre-test.
2.2 Selection of Sampling Units.
Comment 3: Non-probability, convenience, and intentional snowball sampling cannot be used if not a specific cluster of participants is supposed to be researched. Your sampling method is not representative as convenience sampling with snow balling leads to a biased group of participants. As you did not mention that you target with your research a specific group of participants, random sampling would have been needed to show a generally applicable result.
Regarding the fact that the sample was non-probabilistic, this was added as one of the limitations of this study.
We hope that we have dealt with your’ suggestions satisfactorily and made all the adjustments requested, both in form and substance.
Yours sincerely,
On behalf of my co-authors,
Reference added to the manuscript:
- (Albdour and Altarawneh, 2014) Albdour, Ali Abaaas and Ikhlas Altarawneh. 2014. Employee Engagement and Organizational Commitment; Evidence from Jordan. International Journal of Business, 19, 192-212.
- (Robinson et al., 2004) Robinson, Dilys, Sarah Perryman, and Sue Hayday. 2004. The Drivers of Employee Engagement Report 408. Institute for Employment Studies, UK.
Round 3
Reviewer 1 Report
Dear author(s),
Thank you for the revision and improvement of your manuscript, which looks now ready for publication.